# Recent Progress in Finding Binary Systems with the B[e] Phenomenon

Anatoly S. Miroshnichenko [1,2,3,4,*], Sergei V. Zharikov [4,5,6], Nadine Manset [7], Serik A. Khokhlov [3,4], Atilkhan S. Nodyarov [4], Valentina G. Klochkova [6], Stephen Danford [1], Aizhan K. Kuratova [4], Ronald Mennickent [8], S. Drew Chojnowski [9], Ashish Raj [10] and Devendra Bisht [10]

[1] Department of Physics and Astronomy, University of North Carolina at Greensboro, Greensboro, NC 27402, USA
[2] Central Astronomical Observatory of the Russian Academy of Sciences at Pulkovo, 196140 St. Petersburg, Russia
[3] Fesenkov Astrophysical Institute, Observatory, 23, Almaty 050020, Kazakhstan
[4] Faculty of Physics and Technology, Al-Farabi Kazakh National University, Al-Farabi Ave., 71, Almaty 050040, Kazakhstan
[5] Instituto de Astronomía, Universidad Nacional Autónoma de Mexico, AP 106, Ensenada 22800, BC, Mexico
[6] Special Astrophysical Observatory of the Russian Academy of Sciences, Nizhnyj Arkhyz, Zelenchukskiy Region, 369167 Karachai-Cherkessian Republic, Russia
[7] Canada-France-Hawaii Telescope Corporation, 65-1238 Mamalahoa Hwy, Kamuela, HI 96743, USA
[8] Departamento de Astronomía, Universidad de Concepción, Concepción 4030000, Chile
[9] Department of Physics, Montana State University, Bozeman, MT 59717, USA
[10] Indian Centre for Space Physics, 466 Barakhola, Netai Nagar 700099, India
* Correspondence: a_mirosh@uncg.edu

**Abstract:** This paper describes recent studies of the FS CMa-type objects, a group of stars showing the B[e] phenomenon defined in 2007. The objects exhibit strong emission-line spectra with both permitted and forbidden lines suggesting the presence of a B-type star as well as strong IR excesses due to radiation of circumstellar dust. These properties are hard to explain in the framework of the evolution of single stars with luminosities between ∼300 and ∼30,000 L$_\odot$ typical of most B-type stars. We explore the hypothesis that the gaseous-and-dusty envelopes of FS CMa objects are due to either earlier or ongoing mass transfer between the binary system components. It is hard to detect the secondary components in these systems because of veiling and distortions by the circumstellar matter because of the relative faintness of the companions. Nevertheless, we detected regular radial velocity variations of the spectral lines in MWC 728, 3 Pup, and AS 386 and we found absorption lines typical of cool stars in the spectra of MWC 645, AS 174, and several other objects. The diversity of the secondary components in FS CMa objects is discussed in the context of non-conservative binary evolution.

**Keywords:** spectroscopy; binary system; emission-line stars; circumstellar matter

## 1. Introduction

Stars and stellar systems of all masses go through periods when they are surrounded by large amounts of circumstellar (CS) material in the form of gas and dust. These periods have different durations which mostly depend on the objects' masses and orbital separations in multiple systems. The reasons for the presence of the CS matter include but are not limited to formation of stars in molecular clouds, stellar winds, and mass transfer between components in multiple systems. The CS matter processes starlight and becomes partially ionized; that results in the formation of emission lines. Dusty particles can form in the distant and mostly neutral part of the CS region.

Although many phenomena due to the CS matter have been successfully explained by the theory of stellar evolution, some remain puzzling even with the currently available

wealth of data and sophisticated modeling methods. One of them is the B[e] phenomenon which refers to the presence of both permitted and forbidden emission lines in the spectra of mostly B-type stars (effective temperatures, $T_{eff}$ = 9000–30,000 K).

This phenomenon was discovered by Allen and Swings [1], who found 65 Galactic B-type stars with forbidden lines (e.g., [O I], [Fe II], [N II]) and permitted emission lines (e.g., H I, Fe II) in the optical spectra and large IR excesses due to CS dust. These objects were initially called "peculiar Be stars" due to the emission-line spectra similar to those of classical Be stars. However, the latter show much weaker IR excesses which are produced by the CS gas only. The name "B[e] stars" was suggested by P. Conti [2] to highlight the presence of forbidden emission lines and distinguish it from other types of B-type emission-line stars, which show permitted lines only.

Lamers et al. [3] recognized four subgroups of B[e] objects with known evolutionary status: pre-main-sequence Herbig Ae/Be stars, symbiotic binaries, compact Planetary Nebulae, and a small number of supergiants. They confirmed the discoverers' conclusion that this phenomenon appeared in objects at very different evolutionary stages but with similar conditions in the CS environments. Due to diverse evolutionary stages, Lamers et al. [3] also recommended referring to them as "objects with the B[e] phenomenon" rather than simply as "B[e] stars". However, ∼50% of the originally selected B[e] objects did not fit in any stellar group with known evolutionary status; these were called "unclassified".

Studying some unclassified objects with the B[e] phenomenon, Miroshnichenko [4] noticed several features they have in common. In particular:

- Very strong line-emission optical spectra (in most cases, see Figures 1–3) accompanied by free–free and free–bound continuum radiation, which may produce a strong veiling of the underlying star radiation. Most emission line profiles imply a disk-like CS gas distribution (Figure 1).
- A steep decrease in the IR flux at wavelengths $\lambda \geq 10$ μm (see Figure 1 in [4] and Figure 4 in this paper as an illustration) that implies a compact spatial distribution of the dust. This decrease is typically steeper than that in young stars surrounded by larger disks/envelopes of the dust still remaining in their parental clouds (see Figure 3 in [5] for comparison).
- Location not far from the main-sequence but over a wide luminosity range $2.0 \leq \log L/L_\odot \leq 4.5$ (see Figure 5).

Some or all of these features were found in 23 out of 30 original "unclassified" objects. Based on this fact, Miroshnichenko [4] proposed that they be recognized as a separate group of objects with the B[e] phenomenon. These objects are now referred to as "FS CMa-type objects", named after the prototype star for the entire B[e] phenomenon, FS CMa = HD 45677 [6].

The defining group features prompted Miroshnichenko [4] to suggest that a likely explanation for the presence of the large quantities of CS matter in FS CMa objects is the mass-transfer processes in binary systems. This is because the mass range of objects with these fundamental parameters (roughly between 3 and 15 $M_\odot$) and their locations in the Hertzsprung–Russell diagram were inconsistent with the theoretical predictions for mass-loss rates from single stars [7,8].

With the group having been defined, several dozen new members and candidates have been found in various catalogs, such as the IRAS catalog [9], the NOMAD catalog [10], and others. The newly found objects were described in [11–13]. In addition to the afore-mentioned defining features of the group objects, absorption lines of neutral metals have been detected in the spectra of some of them [14,15]. The strengths of these lines suggest the presence of cool components, which are typically $\Delta V \geq 2$ mag fainter than the hot components. The exception is MWC 623 with both components of a visual brightness nearly the same and the only binary system in the FS CMa group that was known from the time of discovery of the B[e] phenomenon [16]. Examples of the absorption-line spectra of cool components are shown in Figure 6. A number of other FS CMa group objects show radial

velocity (RV) or brightness variations possibly due to orbital motion, while other signs of binarity have been detected via spectroastrometry (e.g., [17]).

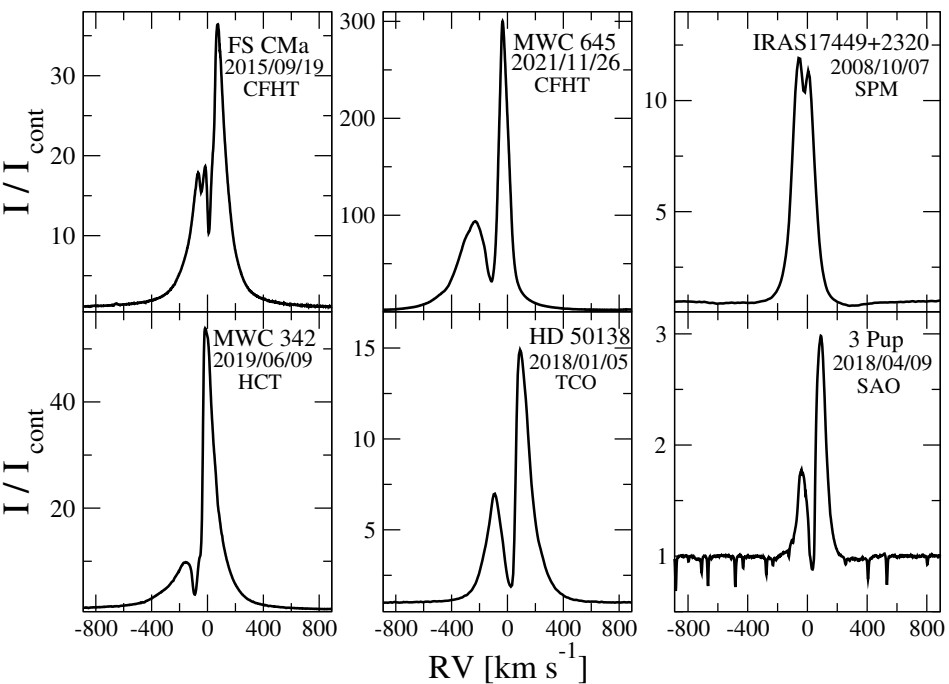

**Figure 1.** Selection of Hα profiles of some FS CMa group objects with strong emission-line spectra. Objects' names, observing dates, and the observatory ID are shown in each panel. Intensity is normalized to the local continuum, and heliocentric RV scale is shown in km s$^{-1}$.

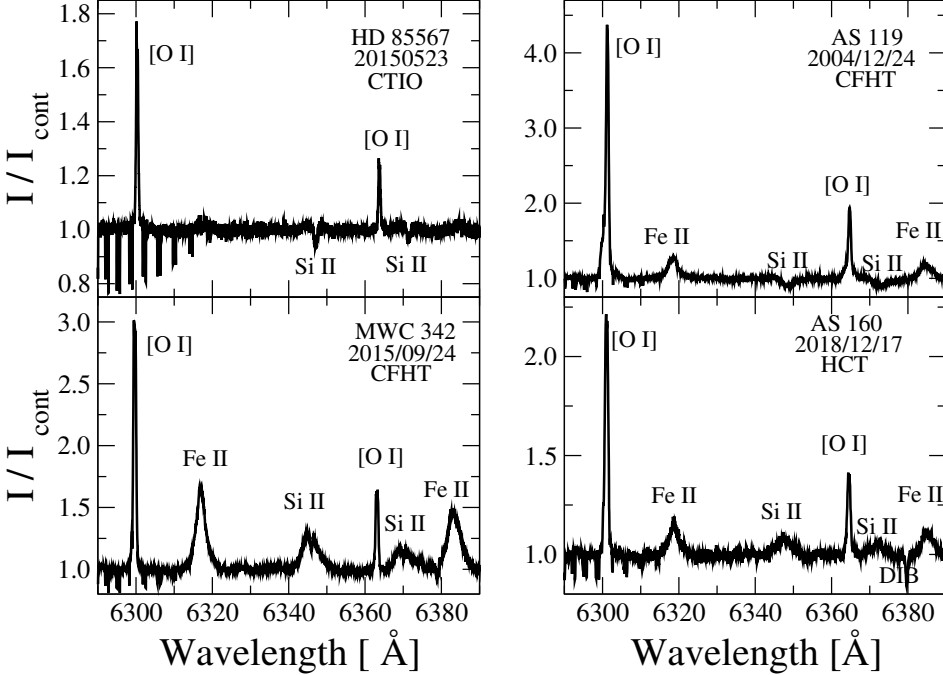

**Figure 2.** Selection of emission-line profiles around the neutral oxygen forbidden lines of some FS CMa group objects. Objects' names, observing dates, and the observatory ID are shown in each panel. Intensity is normalized to the local continuum, and heliocentric wavelength scale is shown in Å. Note: The line at 6318 Å has also been identified as that of Mg I [18] or Mg II [19].

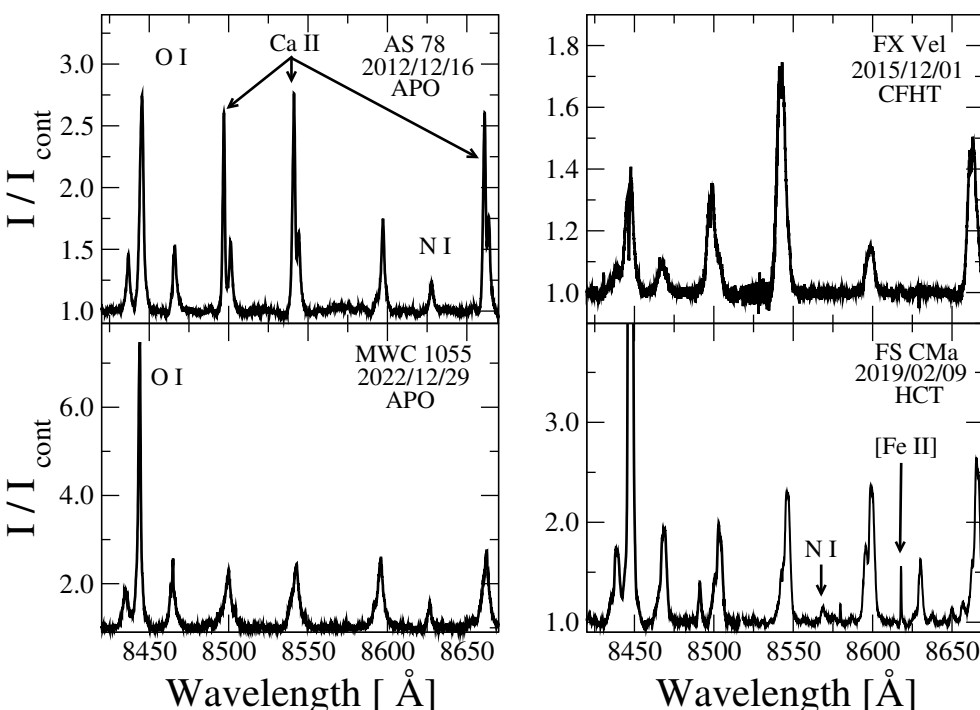

**Figure 3.** Selection of emission-line profiles in the Paschen line region of some FS CMa group objects. Objects' names, observing dates, and the observatory ID are shown in each panel. Most of the lines are those of hydrogen (Pascen series). Lines of other elements are marked. Intensity is normalized to the local continuum, and heliocentric wavelength scale is shown in Å.

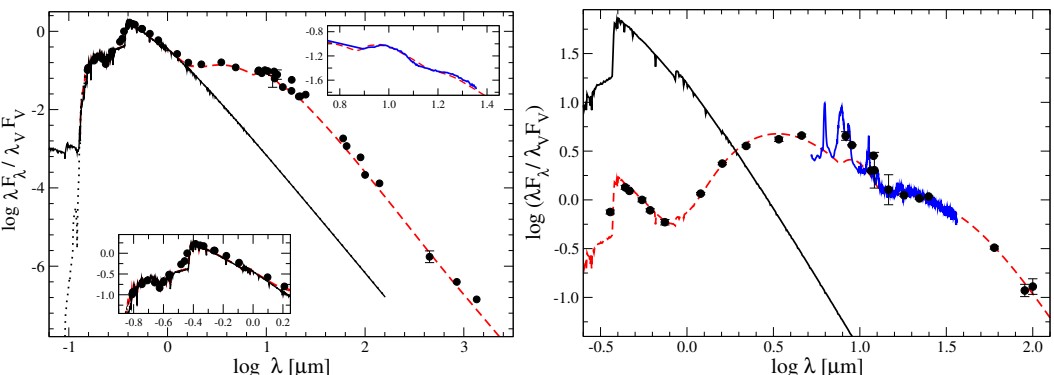

**Figure 4.** Observational and best fitting theoretical SEDs of two FS CMa group objects recently modeled using the radiation transfer code developed by Zakhozhay et al. [20]. The photometric data corrected for the interstellar extinction are shown with black circles. The best fitting theoretical SED are shown with red dashed lines. The fluxes are normalized to that in the *V* band; the wavelengths are shown in microns. Details of the modeling and parameters can be found in [21] for 3 Pup (**left panel**) and in [22] for IRAS 07080 + 0605 (**right panel**). **Left panel:** 3 Pup. Black solid line shows the emergent stellar SED with added flux from the secondary component. The dotted line shows the stellar SED of the primary component only. The upper inset shows the 10 µm region with the IRAS low resolution spectrum (blue line) and its detailed fit. The lower inset shows the agreement of the observed and model UV fluxes. The contribution from the secondary component shows up as a deviation from the black dashed line in the far UV. **Right panel:** IRAS 07080 + 0605. The photometric data and the best fitting model are shown by the same lines and symbols as for 3 Pup. The blue line shows the Spiter Space Observatory spectrum. The emission features near $\lambda \sim 10$ µm produced by Polycyclic Aromatic Hydrocarbons were not attempted in the fitting process. The black line shows the SED of the star without the disk attenuation.

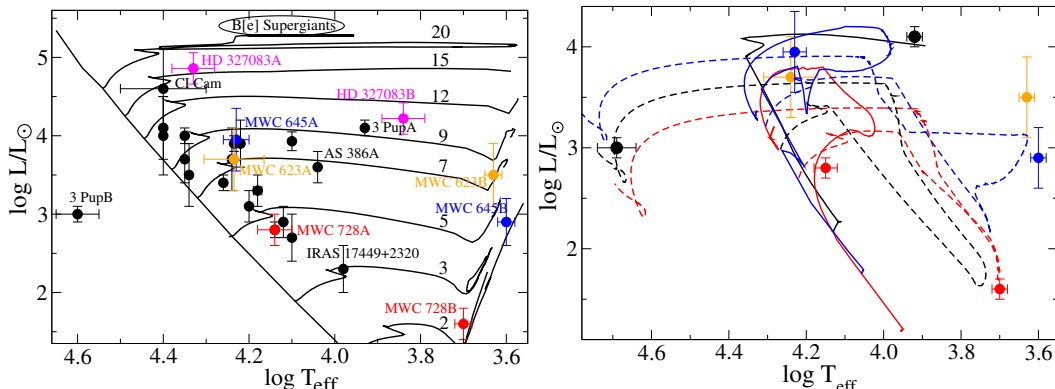

**Figure 5.** Hertzsprung–Russell diagram for FS CMa objects with known parameters. **Left panel:** Recently studied objects are marked with their IDs ("A" stands for the brighter primary component, while "B" stands for the fainter secondary component). Solid lines: the zero-age main-sequence and evolutionary tracks for single rotating stars [23] with initial masses in solar units indicated. **Right panel:** Positions of 4 confirmed double-lined FS CMa-type binaries shown by circles of different colors (same as those in the left panel): 3 Pup (black), MWC 728 (red), MWC 645 (blue), and MWC 623 (orange). Evolutionary tracks of interacting binaries from [24] are shown by the same colors as those of the objects, whose positions are explained by the models. Solid lines show tracks of gainer stars and dashed lines show tracks of donor stars. Initial masses of the shown tracks are as follows: 6.0 M$_\odot$ + 3.6 M$_\odot$—black lines; 5.0 M$_\odot$ + 2.0 M$_\odot$—red lines; 7.0 M$_\odot$ + 2.8 M$_\odot$—blue lines.

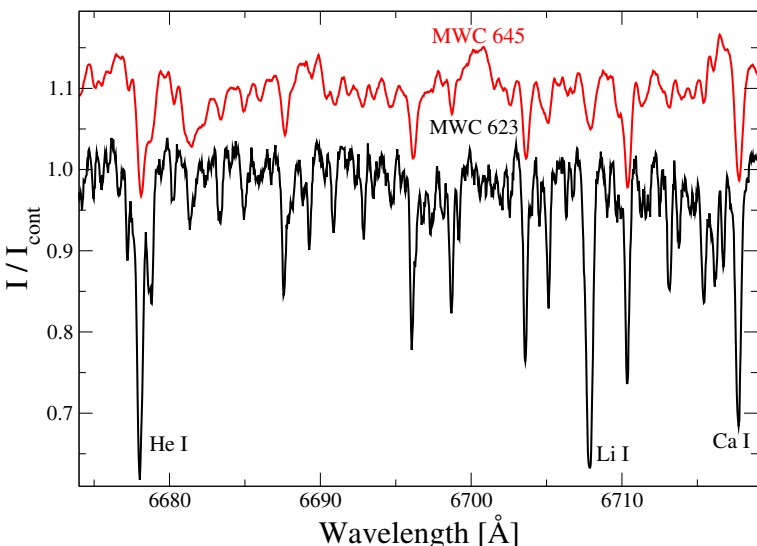

**Figure 6.** Parts of CFHT spectra of two double-lined binaries, MWC 623 (lower) and MWC 645 (upper). The He I 6678 Å lines from the spectrum of the hot component (uncertain for MWC 645 [15]) as well as the Li I 6708 Å and Ca I 6717 Å lines from the spectrum of the cool component are marked. Other absorption lines are mostly Fe I lines from the cool components. Intensity is normalized to the local continuum, and wavelengths are shown in Å. The wavelength scales are shifted to match the line positions of the cool components.

Properties of binary FS CMa objects have been reviewed by Miroshnichenko and Zharikov [25] and Miroshnichenko [26], but more binaries have been discovered since that time. In this paper, we describe these new findings and suggest future directions for studying this group in order to further test the binary hypothesis of its origin.

## 2. Observations

Our observing program mostly includes spectroscopic observations taken at several observatories world-wide. A list of telescopes and instruments where the majority of the spectra have been obtained is presented in Table 1. All the instruments used are échelle spectrographs with a spectral resolving power of $R \geq 12,000$ that allows for measuring precise RVs and resolving most spectral-line profiles. Most spectra were obtained in a wavelength range from $\sim$3800 Å to 7900 Å, although the CFHT, McD, APO, and HESP spectra extend to 10,500 Å. Additionally, we retrieved some spectra taken with the FEROS and UVES spectrographs ($R \sim 40,000$–$57,000$) from the ESO data archive[1].

**Table 1.** Observatories and Instruments.

| Observatory | Telescope | Instrument | Resolution | Location |
|---|---|---|---|---|
| CFHT | 3.6 m | ESPaDOnS | 65,000 | Mauna Kea, Hawaii, USA |
| APO | 3.5 m | ARCES | 31,500 | Apache Point, New Mexico, USA |
| OAN SPM | 2.1 m | REOSC | 18,000 | Baja California, Mexico |
| SAO | 6 m | NES | 60,000 | Nizhniy Arkhyz, Russia |
| McD | 2.7 m | Tull coudé TS2 | 60,000 | Mt. Locke, Texas, USA |
| HCT | 2 m | HESP | 30,000 | Mt. Saraswati, Hanle, India |
| CTIO | 1.5 m | CHIRON | 25,000 & 80,000 | Cerro Tololo, Chile |
| TCO | 0.81 m | eShel | 12,000 | North Carolina, USA |

Column information: 1—Observatory/Telescope acronym; 2—Telescope size; 3—spectrograph name; 4—spectral resolution $R = \lambda/\Delta \lambda$; 5—Observatory location. Acronyms: TCO—Three College Observatory; CFHT—Canada-France-Hawaii Telescope; APO—Apache Point Observatory; OAN SPM—Observatorio Astronómico Nacional San Pedro Martir; McD—McDonald Observatory; SAO—Special Astrophysical Observatory of the Russian Academy of Sciences; HCT—Himalayan Chandra Telescope. Spectrograph references: eShel —https://www.shelyak.com (accessed 15 February 2023); ESPaDOnS—[27]; ARCES—[28]; REOSC—[29]; NES—[30]; TS2—[31]; HESP—[32]; CHIRON—[33].

Our collection contains hundreds of spectra of over 50 FS CMa group members and candidates. Brighter objects ($V \leq 10$ mag) have been observed more frequently, because it is possible to use even relatively small telescopes for them (for example, the 0.81 m telescope at TCO). Two-meter class telescopes (e.g., OAN SPM and HCT) allow one to obtain spectra of $V = 11$–12 mag objects with a higher resolution ($R \sim 20,000$–$30,000$) and reasonable signal-to-noise ratios in the continuum for detecting weak absorption lines and detecting RV variations. Larger telescopes are very important for observing fainter objects, which account for nearly half the entire FS CMa group (see Figure 3 in Miroshnichenko [26]). There is also a subgroup of even fainter objects ($V \sim 15$ mag) with very strong emission lines, such as FBS 0022–021 (e.g., [34]). Observations of these stars are difficult to obtain as they require 5–10 m class telescopes or very long exposure times at smaller telescopes.

To complement the spectroscopic data, multicolor photometry of the FS CMa group objects was collected from publicly available databases in a wide wavelength range from the UV to far-IR. The photometric data combined with the spectral information were used to compose the objects' spectral energy distributions (SEDs) and perform various tasks in the data analysis. In particular, interstellar and total (interstellar plus CS) reddening and extinction were estimated, stellar components' contributions were disentangled wherever possible, and luminosities were calculated based on dereddend SEDs.

Optical and IR photometry of stars projectionally close to the object's locations were used to constrain the interstellar extinction laws (relationships between the interstellar extinction, $A_V$, and distance) in their directions. These relationships are used to verify similar results from large-scale photometric surveys (e.g., [35]), as the latter are unavailable for small distances and crowded regions.

## 3. Results

Early studies (1970s–1990s) of all groups of objects with the B[e] phenomenon from the original list [1] were sparse. More attention was paid to those systems whose nature

and evolutionary status were well understood. At the same time, only a few "unclassified" objects were investigated in detail, such as GG Car [36,37]. Even the prototype star for the B[e] phenomenon, FS CMa, was studied mostly photometrically with much less spectroscopic coverage [38].

Our approach was more focused on spectroscopic observations, although we used every opportunity to obtain multicolor optical and near-IR photometry. As a result, one binary system (V 669 Cep) was discovered before defining of the FS CMa group and nine other binaries have been discovered since then. The latter objects are listed in the lower part of Table 2 (IRAS 07377−2523 and all the objects below it). Several other group members have been suspected in binarity for various reasons listed in Table 3. There are also FS CMa group members with extremely strong emission-line spectra (the Hα line equivalent widths ≥ 500 Å), which suggests that the CS matter is more likely to come from interaction in a binary system but no direct evidence for binarity has been revealed yet. The examples are FBS 0022−021 [11,34], MWC 17 [39].

**Table 2.** Detected or suspected FS CMa-type binary systems.

| Star ID | V | Sp.T. | Period | Ref. |
|---|---|---|---|---|
| MWC 623 | 10.8 | B4 v + K2 iii | − | [16,40] |
| CI Cam | 11.6 | B2 iii + ? | 19.41 ± 0.02 | [41] |
| V 669 Cep | 12.2 | B4/6 + lt | − | [42] |
| HD 50138 | 6.6 | B5/6 V | − | [17] |
| FS CMa | 7.0–8.9 | B2 v | − | [17] |
| HD 85567 | 8.6 | B5 v | − | [17,43] |
| IRAS 07377−2523 | 12.8 | B8/A0 + lt | − | [11] |
| FX Vel | 9.4–11.4 | A + lt | − | [11] |
| AS 174 | 11.5 | A + lt | − | [11] |
| IRAS 00470 + 6429 | 12.0 | B2 v + lt? | − | [44] |
| MWC 728 | 9.8 | B5 v + G8 iii | 27.50 ± 0.11 | [14] |
| AS 386 | 10.9 | B9 Ib + ? | 131.27 ± 0.09 | [45] |
| 3 Pup | 4.0 | A3 Ib + sd O | 137.4 ± 0.1 | [21] |
| MWC 645 | 12.8–13.4 | B2 v + K2 iii | − | [15] |
| HD 327083 | 9.7–10.1 | B1 I + F1 I | 107.68 ± 0.02 | [46,47] |

Column information: 1—the object identification; 2—average brightness or the range of brightness variations of the entire system in the *V*-band; 3—MK types of the system components ("lt" stands for "late-type"); 4—orbital period in days (where available); 5—reference to the paper(s), where binary status was discovered or somehow discussed. HD 50138, FS CMa, and HD 85567—binaries detected by spectroastrometry (see Section 4.4). The *V*-band brightness of CI Cam is shown in quiescence.

Our early discoveries were based on a few spectra for each object and only revealed those with composite spectra in relatively bright ones ($V \leq 12$ mag). Fainter ones required higher signal-to-noise ratios. This was recently achieved for a 13-mag object MWC 645, in whose CFHT spectra absorption lines of a cool component were detected with exposure times of several hours. Adding more spectra for a few other objects allowed us to determine orbital parameters for single-lined systems MWC 728, AS 386, 3 Pup, and a double-lined one HD 327083. Preliminary parameters of the latter binary, which shows absorption lines from the cool component and emission lines from the hot one, found in [46] with eight spectra were refined by doubling the number of them. The orbital period of the brightest object with the B[e] phenomenon, 3 Pup, derived for the first time nearly 70 years ago [48] was confirmed in [21] with over 100 spectra taken at the 0.81 m TCO telescope. This large number of new observations allowed the determination of the RV amplitude for the first time and thus constrained the properties of the secondary component.

Orbital periods in all the above-mentioned FS CMa objects were derived from analysis of absorption-line RVs using at least 20 high-resolution spectra with signal-to-noise ratios of ≥50 in the continuum. Photometric variations corresponding to the spectroscopic period have been found for HD 327083. The orbital period in the CI Cam system was found from the photometric data and confirmed with the RV variations of only one emission line (He ii 4686 Å).

**Table 3.** Candidate FS CMa-type binary systems.

| Star ID | V | Sp.T. | Ref. | Comment |
|---|---|---|---|---|
| MWC 342 | 10.2–10.9 | B2 v | [49] | several brightness variation periods & X-ray source near the star position |
| AS 160 | 10.9 | B1 v | [50] | RV difference between absorption and emission lines |
| AS 119 | 12.5 | B3 v | [51] | RV difference between absorption and emission lines |
| VES 723 | 13.2 | B0 v | [52] | emission-line profile shape suggestive of ongoing mass transfer |
| IRAS 17449 + 2320 ⋆ | 10.0 | B9 v | [53] | periodic appearance of an absorption features in many lines |
| IRAS 07080 + 0605 | 11.9–12.2 | A3 v | [22] | absorption line RV variations detected but unconfirmed due to an insufficient amount of data |

Column information: 1—the object identification; 2—*V*-band brightness range; 3—MK type of the star; 4—reference to the paper; 5—reason(s) for suspecting binarity. ⋆—the periodicity will be described in detail by Miroshnichenko et al. (2023, in preparation).

## 4. Discussion

The binaries with known properties of both components show a clear dichotomy. One group of them has cool secondary components, while the secondary component in the other group is either very hot dwarf (stripped stellar cores) or a degenerate remnant of stellar evolution such as a white dwarf, neutron star, or black hole. The nature of the secondaries of the latter group is deduced either from the mass function or from phenomenological arguments. Fundamental parameters of the FS CMa objects derived via various methods (e.g., kinematical distance, Gaia parallax, or interstellar extinction law to derive luminosity and spectral line properties together with SEDs to derive $T_{eff}$) are shown in Figure 5. Known binaries with a variety of secondaries are considered below.

### 4.1. Late-Type Secondaries

There are eight FS CMa-type objects in which absorption lines of neutral metals have been detected (see Table 2) indicating the presence of a cool secondary component. In all of these cases, the strength of the neutral metal lines indicates that the hot components are brighter than the cool ones.

#### 4.1.1. MWC 623

MWC 623, an object from the original list of B[e] stars [1], shows the strongest absorption-line spectrum in the entire group (see Figure 6) along with narrow emission lines and very small RV variations despite many years of spectroscopic observations. Its spectrum was analyzed in [16,54], where the components' properties were estimated. However, this has not been studied from the standpoint of the interacting binary evolution. Future efforts need to focus on a more careful disentangling of the components' contributions to the total observed flux, including emission-line profile modeling and new optical broad-band polarization measurements.

The small RV variations have been interpreted as either due to a large orbital separation between the components or to a nearly pole-on orientation of the binary and its CS disk [16,55]. The latter explanation seems to be less probable, as the system shows a high intrinsic polarization inconsistent with a high inclination angle [56]. Polster et al. [55] suggested that the entire absorption-line spectrum of MWC 623 might form in an optically thick equatorial disk viewed nearly edge-on, which acts as a pseudo-photosphere. These authors only applied this model to possible kinematic effects on the spectral line positions and concluded that it was still consistent with the object's binarity, even if the absorption lines were formed in the disk.

#### 4.1.2. MWC 728

The first FS CMa object with a cool secondary component and a measured orbital period was MWC 728 [14]. The contribution of the secondary was found to be 10% to

the total observed brightness in the *V* band which made the absorption-line spectrum weak. The H$\alpha$ line profile is generally double-peaked but shows an additional weak central emission peak, which may originate in the ionized material transferring between the components. Comparison with evolutionary models of close binary systems [24] shows that the cool component began its evolution as a 5 M$_\odot$ star, while the B-type primary was a 2 M$_\odot$ star. The mass ratio is now reversed and the mass transfer should be still ongoing. According to this model, up to 2 M$_\odot$ is going to be lost from the system when the mass-transfer process stops. This prediction is consistent with the large strength of the emission-line spectrum, which is typically much weaker in Be stars of the same spectral type (B5) and which cannot be explained by a single star evolutionary stellar wind. There is a contradiction between the distance to the system based on the interstellar extinction law (1.0 kpc) adopted in [14] and the most recent Gaia measurement (0.3 kpc [57]).

This contradiction may be due to the complexity of the systems (at least two stars plus a gaseous and dusty envelope, which re-emits and scatters the starlight), which makes parallax measurements difficult. It is also not the only case of such a contradiction among the FS CMa group (see below).

### 4.1.3. HD 327083

The third object in this category is HD 327083, whose emission-line spectrum was found long ago [58] and which was overlooked by Allen and Swings [1] despite its brightness in the near-IR region (*K*–band brightness = 3.3 mag). The reason is probably a strong interstellar/CS reddening that masked the IR-excess due to the CS dust radiation. Initial studies strongly suggested that its properties are dominated by the existence of an early B-type star. Machado et al. [59] and Machado and de Araújo [60] concluded that it was a single extremely luminous supergiant (log L/L$_\odot$ $\sim$ 6.0), while Miroshnichenko et al. [46] found absorption lines of a cooler component in its spectrum and detected strong variations of both emission and absorption lines in opposite directions. At the same time, the orbital period was determined incorrectly due to a small number of spectra used in [46].

When photometric monitoring by the ASAS programs [61,62] became available, Maravelias et al. [63] found several periodicities in the brightness variations of the object but they have not studied their origin. Using a larger collection of high-resolution spectra, Nodyarov et al. [47] found that a 107.7-day period is present in both photometric and spectroscopic variations. These authors confirmed the conclusion from [46] about the components' T$_{\text{eff}}$ ($\sim$20,000 K and $\sim$7000 K) and determined their luminosities (log L/L$_\odot$ $\sim$ 5.0 and $\sim$ 4.5 for the hot and cool component, respectively) using the latest Gaia distance [57]. Comparison with evolutionary tracks for single stars [23] suggests an initial mass of $\sim$15 M$_\odot$ for the hot component and $\sim$10 M$_\odot$ for the cool one (see left panel of Figure 5). These estimates are much larger than those found from the orbital solutions ($\sim$7.5 M$_\odot$ for each of the components) using the mass functions and the system inclination angle (60°) derived from the light curve modeling. The discrepancy may be due to a distance determination error and/or an ongoing mass transfer, which breaks the stellar structure equilibrium and thus alters the mass–luminosity relationship.

### 4.1.4. Other Hot/Cool Binaries

Five other FS CMa group objects (V 669 Cep, FX Vel, AS 174, IRAS 00470 + 6429, and MWC 645) show spectral lines of supposedly cool components. Positional changes of the cool component's lines were only established in the case of MWC 645 [15], but the orbital period was not determined because the object is faint and only two spectra were suitable for the RV measurements. The other four objects either exhibit very few such lines (e.g., IRAS 00470 + 6429 only shows the Li I 6708 Å line [64]) or the number of spectra was insufficient to conclude much regarding changes in the positions of the lines. Further progress in studies of these objects is expected soon, as more spectra of all of them have been taken since the the first studies (2007–2009).

The presence of absorption lines of neutral metals was not the only observation explained by the presence of a cool secondary component. Miroshnichenko et al. [64] suggested that the Li I line might have been due to the "Hot Bottom Burning" process at the post-AGB evolutionary stage (e.g., [65]), but they found that the luminosity of IRAS 00470 + 6429 was not high enough to run this process. In Section 4.1.1 we also mentioned the idea by Polster et al. [55] that these lines may form in the optically thick CS disk.

### 4.2. Hot Stripped-Core or Degenerate Secondaries

#### 4.2.1. CI Cam

The binarity of CI Cam was initially proposed by Miroshnichenko [66], who found absorption lines typical of a cool star in the optical spectrum. However, these lines turned out to be due to instrumental effects rather than real lines. A strong outburst of the object occurred on 31 March/1 April 1998 and was detected in a very large wavelength range from X-rays to radio wavelengths. This event received a variety of interpretations. In particular, Belloni et al. [67] suggested that the secondary companion is a black hole or a neutron star, while Orlandini et al. [68] speculated that it was most probably a white dwarf. Barsukova et al. [41] found a 19.4-day period in the *V*-band brightness variations and the RV variations of a broad He II 4686 Å emission line, which is supposed to be formed in the accretion disk of a white dwarf. The nature of the primary component has not been constrained until recently due to a distance uncertainty (between 1 and 10 kpc) and the absence of photospheric lines in its spectrum. However, both ground-based and TESS photometry revealed pulsations with several periods. The main pulsation period that has been observed since 2018 is 0.4062 days which is interpreted as a resonance of radial modes by Barsukova et al. [69]. The latter fact constrains the MK-type of the primary component as B0–B2 III, rejects its supergiant classification, and suggests that the secondary component is most likely a $\sim 1$ M$_\odot$ white dwarf. However, it could also be a stripped core of an initially more massive star. Nevertheless, these new results place the CI Cam system in the FS CMa group.

#### 4.2.2. 3 Pup

The 3 Pup (HR 2996) is similar to CI Cam and was included in the original list of B[e] stars. It is the brightest object with the B[e] phenomenon (*V* = 3.96 mag). Spectroscopically, the star has always been classified as an early A-type supergiant (e.g., A2.7 Ib [70]), but its luminosity, $\log L/L_\odot \sim 4.0$, is too low to include it in the B[e] supergiant group, whose average luminosity was suggested to be $\log L/L_\odot = 5.1 \pm 0.2$ [4]. A small number of photographic and photoelectric spectra obtained for the object in the 20th century allowed detection of RV variations of absorption lines with a period of 137 or 161 days [48,71].

However, the amplitude of the RV variations was not constrained until an extensive monitoring performed by Miroshnichenko et al. [21], who measured the semi-amplitude to be $5.0 \pm 0.8$ km s$^{-1}$. This result suggested that the secondary component is a low-mass star, whose contribution to the visible portion of the spectrum was undetectable. These authors suggested that the secondary is a stripped core remnant of an initially more massive component and proposed an evolutionary history of this 6.0 M$_\odot$ + 3.6 M$_\odot$ system. Their conclusion was based on models calculated using the code described in [24]. The current components' masses are 8.8 M$_\odot$ and 0.8 M$_\odot$.

The secondary component has $T_{\rm eff} \sim 50{,}000$ K and only dominates the total flux of the system at wavelengths shorter than $\sim 1000$ Å, where no observations of the object exist (see left panel of Figure 5). The evolutionary model suggests a fully conservative mass transfer, although CS matter is present in the system in the form of ionized gas (emission lines) and dust (IR excess). This matter is most likely a result of the mass transfer in the past. However, the amount of this gaseous and dusty material is $\sim$0.01 M$_\odot$ and may be too small for the current version of the evolutionary modeling codes to be considered a non-conservative mass loss.

### 4.2.3. AS 386

The only candidate for a binary system with a high-mass degenerate component in the FS CMa group is AS 386 [45]. Its optical spectrum contains many absorption lines typical of a late B-type supergiant (B9 Ib) but also an excessive number of lines of such elements as silicon, neon, and aluminum, which are unusual in the spectra of such normal stars. Its emission lines are not very strong but exhibit regular variations in the peak intensities (violet over red, V/R) of double-peaked profiles with the same period (131.4 days) as that of the absorption-line positions. The same period is observed in the near-IR brightness variations. The orbital parameters and the absence of any observable signs of the secondary component were interpreted in [45] as the presence of a $\geq 7\,M_\odot$ black hole in the system.

However, there is another possible way to form a binary system like AS 386. If the mass-transfer process ended recently, the visible star in this system (mass donor) has not yet restructured and may have a much smaller mass than that derived from the evolutionary tracks of single stars while still having a high luminosity. As a consequence, the secondary component may still be a "normal" B-type star with a large rotational velocity and very shallow absorption lines, which are very difficult to detect. A similar analysis was applied in [72] to HR 6819, which was suggested to be a triple system with a possible black hole in [73]. As a result, the authors of [72] argue that HR 6819 is a Be + B binary system. The conclusion that formation of binary systems with a black hole is unlikely was also reached in [74]. UV observations are required to verify whether a fainter hot star is present in such systems.

### 4.3. Suspected but Unconfirmed Binaries

Detecting the effects of binarity is not easy in FS CMa objects due to processes in the CS medium which are simultaneous with orbital variations and may completely hide the latter. This complexity calls for a multi-technique long-term monitoring that sometimes requires decades of observations. However, even this approach does not always lead to conclusive results.

We mentioned above the story of discovering binarity of the brightest group object, 3 Pup (see Section 4.2.2), which took $\sim$75 years before the orbital solution was established. This story is not over yet for the next three brightest group objects (HD 50138, FS CMa, and HD 85567). There is still a controversy about the evolutionary classification of all of them. In some studies they are considered pre-main-sequence Herbig Be stars (e.g., [75–77] to name a few recent ones), but there has not been an unambiguous conclusion as to their extreme youth. Neither of these objects belongs to a star-forming region or has a strong far-IR excess from distant CS dust radiation that is similar to those of true pre-main-sequence stars. Recently, Rich et al. [76] claimed detection of scattered light at distances over 100 AU from these objects which was regarded as "a sign of the extreme youth", but their final conclusion was only that "these objects are likely young systems".

No regular periodic variations have been found in their spectra (e.g., [43,78]), but signs of binarity were detected in spectroastrometric observations of all three [17]. This method uses long-slit spectroscopy with two orthogonal orientations of the slit. Spectra obtained in this way are analyzed for the position of the continuum and emission lines on the detector. A difference between the two is interpreted as resulting from a different gravitational attraction to the secondary component. This result is qualitative and does not provide any orbital parameters. The method was successfully applied to many binary systems, including Be and pre-main-sequence stars. Its use allowed the discovery of previously unknown binaries (e.g., [79]). Even though the positional differences between the continuum and the H$\alpha$ line in the spectra of all the three objects was clearly detected, their binarity is usually not acknowledged or used in other studies.

Finally, various regular or quasi-regular variations have been detected in the six objects listed in Table 3 and a few more (e.g., FS CMa and AS 78, where cyclic behavior has been suspected but the studies are still in progress). They are all due further careful

study. In the following paragraphs, we describe some known details of the most promising binary candidates.

Recently, Korčáková et al. [53] found split absorption lines in the optical spectrum of IRAS 17449 + 2320 and interpreted them as due to Zeeman effect from a strong magnetic field (6.7 KGauss). This is surprising, because typically B-type emission-line stars exhibit barely detectable magnetic fields (e.g., [80]). Moreover, magnetic Ap stars show a strong decrease in their magnetic fields to $\leq$1000 KGauss by the end of their main-sequence evolution [81], where IRAS 17449 + 2320 is found (see left panel of Figure 5).

A subset of high-resolution spectra from over 100 spectra used in this study showed a periodic onset of a strong absorption feature at the red edge of many spectral lines (e.g., Balmer series, oxygen, helium, etc.) with a period of 36.13 days [82], which may be the star's rotation period as suggested in [53]. We are still studying this phenomenon and will draw conclusions on the object's binarity soon.

Another unexpected result was obtained for IRAS 07080 + 0605 [22], whose properties turned out to be very similar to those of the well studied proto-planetary nebula and a binary system Red Rectangle (HD 44179). The major difference between the two objects is that the former shows no visible nebula, despite being closer to the Sun than the latter. Similarly to Red Rectangle, IRAS 07080 + 0605 is viewed through an optically thick dusty disk which attenuates the optical flux from the central star by a factor of ∼40 in the *V* band (see right panel of Figure 5).

Our collection of ∼30 high-resolution spectra shows a number of absorption lines whose positions vary with an amplitude of ∼20 km s$^{-1}$ and a period of ∼24 days. However, the spectra were obtained over a long period of time (17 years), and such a short period may not be real. Moreover, the object shows cyclic optical photometric variations with a very different characteristic period of ∼190 days. Both regular photometric and spectroscopic variations have been detected in Red Rectangle, but their periods are the same (∼320 days) within an uncertainty of a few days (see, e.g., [83]). We are planning a monitoring campaign to obtain more spectra of IRAS 07080 + 0605 to follow short-term spectral variations.

*4.4. Other Problems*

Although binarity does not rule out a young age, the mechanisms of the CS matter formation in evolved interacting binary systems are different from those in young systems. While pre-main-sequence stars accrete the material from typically large parental clouds and thus naturally have distant CS matter, the mass-transfer process in evolved binaries forms it from inside out which makes it typically more compact. Despite the discovery of distant scattered light in FS CMa, HD 50138, and HD 85567 through near-IR polarimetry, their SEDs show a steep decrease of the flux longward of $\lambda \sim 10$ μm that is atypical among pre-main-sequence objects. The latter implies the lack of the cold dust in their CS area which we interpret as a sign of an advanced evolutionary stage. The scattered light may indeed be due to the remnant of the parental clouds, but this is typical around even main-sequence stars, such as Vega [84].

Moreover, there are isolated pre-main-sequence stars (not connected to large star-forming regions), but they typically either still exhibit optical nebulae or have relatively low masses, i.e., evolve slower than more massive stars and are incapable of dispersing debris of their clouds quickly due to weaker stellar winds. The three FS CMa objects in question seems to be in the middle of the Herbig Be star mass range (4–8 M$_\odot$) and should have a very short evolutionary time before starting on the main sequence. All these arguments lead to doubts about their very young ages.

There is also a problem with the luminosity determination for some FS CMa objects for the following reasons. First, it is not easy to account for the effect of CS matter as it both absorbs the radiation of the underlying stars and re-emits their ionizing UV radiation in spectral lines and optical and IR continuum. Second, distances toward them determined by different methods are inconsistent with each other in a number of cases. As was mentioned in Section 4.1.2, the Gaia distance [57] for MWC 728 is much smaller than that determined

from the interstellar extinction law. At the same time, the Gaia distance to AS 386 is nearly twice as large compared to that derived in [45]. Hopefully more Gaia orbits will resolve these issues.

### 4.5. Evolution of FS CMa Binaries

The properties of FS CMa objects (emission-line spectra, IR excesses, variability) suggest that they undergo different stages of evolution. For example, stronger emission-line spectra and IR excesses imply a larger amount of CS matter and indirectly imply a more active stage, which is most likely due to an ongoing mass transfer in the system. Alternatively, weaker emission-line spectra may indicate that the mass-transfer stage is over. However, more certain conclusions can be deduced from the binaries with available information about both components.

The right panel of Figure 5 shows fundamental parameters of the four double-lined FS CMa binaries along with evolutionary tracks calculated for various combinations of the components' initial masses. Only the model for 3 Pup closely reproduces the system's current parameters (components' masses and orbital period). The models for MWC 728 and MWC 645 match the component's fundamental parameters with a good accuracy, but the observed orbital period for MWC 728 is somewhat larger than that predicted by the model and the orbital period for MWC 645 is not even known. MWC 623 has not been studied as an interacting binary.

One difficulty with finding the right evolutionary model is small sizes of the published grids, while another one is the small number of FS CMa binaries with known orbital periods. Nevertheless, even the existing data allow us to propose possible explanations for the observed properties of the objects in the FS CMa group.

The first mass transfer in the system due to the Roche lobe overflow (RLOF) by the initially more massive component begins after it finishes the main-sequence evolutionary stage. As a consequence, its luminosity and $T_{eff}$ begin to decrease. Simultaneously, the less massive component accelerates its evolution as it gains mass; its luminosity increases and $T_{eff}$ begins to increase. This stage ends when the donor becomes the less massive component, finishes its loop in the HR diagram and continues the evolution at a nearly constant luminosity.

Therefore, the three systems with the cool components, whose evolutionary models are shown in the right panel of Figure 5, are still at the RLOF stage. Consequently, all these objects show strong emission-line spectra. The emission lines are weaker in MWC 728 because the gainer in this system has a lower mass and a lower $T_{eff}$ compared to the others; this makes its ionizing power lower.

At the same time, 3 Pup seems to be at the post-mass-transfer stage with a relatively weak emission-line spectrum due to the remaining CS matter in the gaseous disk and stellar wind from the gainer. A similar situation seems to be occurring in the AS 386 system irrespectively of the nature of its secondary (donor) component and in HD 327083, whose emission-line spectrum is weak and the IR excess is small. All these systems have orbital periods over 100 days, and none of their components fills their Roche lobes. CI Cam has a strong emission-line spectrum, which may be explained by a smaller distance between the components (orbital period 19.4 days) that provides conditions for mass transfer from the former gainer back to the currently less massive secondary. Moreover, this is the only eccentric system ($e$ = 0.44–0.49 [69]) among the FS CMa binaries with determined orbits.

The other FS CMa group members shown in the left panel of Figure 5 are located in the same area as the B-type (gainers) components of the systems with cool secondaries. This suggests that the mass transfer process is not over in them, as they show strong emission-line spectra. One may expect orbital periods in these systems on the order of a few weeks. However, some of them may be products of mergers.

## 5. Conclusions

We described the current state of investigation of the FS CMa group of objects showing the B[e] phenomenon with the emphasis on recent results. The main hypothesis intended to explain the strong emission-line spectra of most of them along with IR excesses, which show a steep decrease of the flux at $\lambda \geq 10$–$30$ μm, involves an ongoing or past mass transfer in close binary systems. Nearly half of this relatively large group, which contains ∼70 objects and candidates, show signs of binarity or secondary components have been detected in them.

Analysis of the fundamental parameters, which have been determined for ∼20 objects, shows that some revealed binaries still undergo a mass transfer process, while some others have already finished it. The orbital periods range from ∼ 3 weeks (CI Cam) to ∼4 months (3 Pup and AS 386). Although some of the group members may be products of mergers, no evidence for this case has been found so far.

On the observational side, we recommend long-term spectroscopic monitoring as the main method of revealing the nature of these objects. Photometric observations are also important, but they are available through regular optical sky surveys, such as ASAS SN and TESS, which allow studying the objects at various time scales from days to years. Near-IR photometry is also helpful, as it traces the CS dust and may reveal regular brightness variations that are veiled in the optical region by a more abundant CS gas. Observations in the far-UV spectral region may reveal the presence of very hot secondary components (e.g., stripped cores of former mass donors). Our spectral database is still growing, and we expect to complete the analysis of more group objects within the next few years.

Observations at higher spatial resolution, such as IR interferometry, have been obtained for a few brightest objects of the FS CMa group (e.g., FS CMa [85], HD 50138 [86], 3 Pup [87]) and resulted in constraining the CS disk structure and parameters. However, neither allowed one to draw an unambiguous conclusion concerning the objects' evolutionary status or the binarity. Complementing spectroscopy and photometry with interferometry at optical wavelengths (e.g., with the CHARA array) may help in revealing more properties of the known binaries and help in resolving suspected ones, especially those at smaller distances (e.g., HD 50138, FS CMa, MWC 728, MWC 342).

On the data analysis side, it is important to model emission-line profiles along with IR excesses in order to constrain the amount and spatial distribution of the CS matter (both gaseous and dusty). These results should allow us to check out the non-conservative models of binary evolution and eventually develop methods of predicting the evolution of the CS matter in such systems. In particular, emission-line profile modeling should constrain the CS geometry and fractional contribution of the disk and wind material in their formation. Only a few studies that involve such a modeling of Balmer emission line profiles have been published for FS CMa-type objects (e.g., [44,88]). Moreover, it is worthwhile mentioning modeling of forbidden emission lines of some objects with the B[e] phenomenon that included a few FS CMa-type objects by Zickgraf [19]. This paper presented a mathematical approach to modeling these lines which has not been applied since that time.

FS CMa objects may be a noticeable source of planet building material in galaxies, as they produce CS dust that later becomes interstellar, and the group is larger than, for example, the group of dust-producing Wolf–Rayet stars (e.g., [89]), which has been considered non-negligible contributors dust producers in the Galaxy [90]. So far dusty environments have been modeled only in a few FS CMa objects. In particular, the SED of FS CMa (HD 45677) was modeled long ago with a spherical dust distribution [91], which is inconsistent with the double-peaked emission line profiles typical for gaseous disks and IR interferometric data consistent with a disk-like shape of the CS dust [85].

More recent modeling results have been obtained for IRAS 00470 + 6429 [44] using the code *HDUST* [92] and 3 Pup with a disk-like silicate dust distribution [21] as well as IRAS 07080 + 0605 with a disk-like carbon dust distribution [22] (both of these results were obtained with the code from [20] and are shown in Figure 5). Repeating this modeling

with other codes and expanding it to more objects would allow a statistical study of the CS dust properties in FS CMa objects and comparison with those of other groups showing dusty environments (e.g., Herbig Ae/Be stars and post-AGB dusty binaries). Most FS CMa objects have been observed in the IR region up to $\lambda \sim 100$ µm, thus allowing for reliable modeling, comparing the results from different codes, and conclusions on the efficiency of the dust production.

**Author Contributions:** Observations, A.S.M., S.D., N.M., S.V.Z., A.R.; Data reduction, A.S.M., N.M., S.V.Z., A.R.; Data analysis, A.S.M., S.V.Z., A.S.N., S.A.K.; Software N.M. and S.V.Z.; writing—original draft preparation A.S.M.; writing—review and editing A.S.M., S.D., S.D.C., R.M., N.M., S.V.Z., V.G.K., A.K.K., D.B. and A.R. All authors have read and agreed to the published version of the manuscript.

**Funding:** This research was funded in part by the Science Committee of the Ministry of Education and Science of the Republic of Kazakhstan (Grant No. AP19578879). S.V.Z. acknowledges PAPIIT grants IN102120 and IN119323. V.G.K. thanks the Russian Science Foundation for support (grant No. 22-12- 00069). R.M. gratefully acknowledges support by the ANID BASAL projects ACE210002 and FB210003 and FONDECYT Regular 1190621.

**Data Availability Statement:** CFHT data are available online from http://polarbase.irap.omp.eu/ (accessed on 15 February 2023). Other original spectra reported in this study are available on request to the first author via email at a_mirosh@uncg.edu.

**Acknowledgments:** This research has made use of the SIMBAD database, operated at CDS, Strasbourg, France; SAO/NASA ADS, ASAS, and Gaia data products. This paper is partly based on observations obtained at the 3.6 m Canada–France–Hawaii Telescope (CFHT), which is operated by the National Research Council of Canada, the Institut National des Sciences de l'Univers of the Centre National de la Recherche Scientifique de France, and the University of Hawaii, as well as on observations obtained at the 6 m telescope of the Russian Academy of Sciences, 2.7 m Harlan J. Smith telescope of the McDonald Observatory (Texas, USA), the 2.1 m telescope of the Observatorio Astronómico Nacional San Pedro Martir (Baja California, Mexico), and 2 m Himalayan Chandra Telescope at the Indian Astronomical Obsevatory (Hanle, India). The observations at the Canada–France–Hawaii Telescope were performed with care and respect from the summit of Maunakea, which is a significant cultural and historic site. The UNCG team acknowledges technical support from Dan Gray (Sidereal Technology company), Joshua Haislip (University of North Carolina Chapel Hill), and Mike Shelton (University of North Carolina Greensboro) as well as the Three College Observatory funding by the College of Arts and Sciences and Department of Physics and Astronomy of the University of North Carolina Greensboro.

**Conflicts of Interest:** The authors declare no conflict of interest.

### Abbreviations

The following abbreviations are used in this manuscript:

| | |
|---|---|
| RV | radial velocity |
| R | spectral resolving power |
| TCO | Three College Observatory |
| RLOF | Roche Lobe Overflow |
| SED | spectral energy distribution |

## Note

1. Accessible at https://eso.org/USERPortal/, accessed on 31 December 2022.

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
