# Peer review of "Recent Progress in Finding Binary Systems with the B[e] Phenomenon"

_galaxies, doi:10.3390/galaxies11010036_

Round 1
Reviewer 1 Report
see document

Author Response
See attached MS Word document.

Reviewer 2 Report
This is a well written and interesting paper. I was aware of this class of objects, but not some of the more recent developments. They are a fascinating and quite a complex and diverse class of objects. The paper has a good number of references and is a very useful review of the topic.
One query, not directly related to the paper - has anyone used the large spectral databases (SDSS/LAMOST) to do a systematic search for these sorts of objects, down to fainter magnitudes?
I note only a couple of small typographical things to be fixed before
publication - listed below.
1) Small point - you don't give the units of the period in Table 2. I know it is in days, but for binaries in general it could be minutes, hours or years.
2) Sect 4.5 you have Teff - should be T_{eff} - it is incorrect in a few places
3) p.14 "a a mathematical" - extra "a"
4| Acknowledgements:
p.15 "technical support FROM" (a word missing I think)
p.15 "aw well" --> "as well"
There may be other typos, so the authors may want to give the paper one more
read through.
Author Response
Please see the attachement
